# TELL, DON'T SHOW: INTERNALIZED REASONING INFLUENCES HOW LLMS GENERALIZE

## ABSTRACT

In this paper we investigate to what extent language models' generalization behavior during a domain shift can be influenced by declarative knowledge contained in the training data. In order to study this we finetune language models to fit some distribution which has a "natural" generalization when the distribution shifts. We then test to what extent declarative statements in the training data - that if fully internalized would greatly affect the domain shift generalization - can indeed alter the model's behavior on unseen examples. While the effect is subtle, the declarative knowledge provided in the finetuning sets systematically changes the models' predictions in the way one would expect. Evidence for the strength of this effect growing with model size is mixed. We further show that the effect can not be explained by simple token matching behavior as it persists even when there is no overlap between the declarative descriptions and the models' test time generations.

## 1 INTRODUCTION

Recently, large language models (LLMs) have attracted significant attention due to their rapidly improving capabilities as seen with models such as GPT-4 (OpenAI, 2023), LLaMa-2 (Touvron et al., 2023), Claude 2 (Anthropic, 2023) and Falcon (Penedo et al., 2023). As they are being used for a growing number of applications, it becomes more important to understand how their training data determines their generalization on unseen examples. In particular, when an LLM is presented with a novel input, it is still debated whether they are merely repeating low-level statistical patterns to match the next token ("stochastic parrot") (Bender et al., 2021) or whether they are capable of using higher-level reasoning for the generation of their outputs. Understanding the mechanisms of LLM generalization is crucially important to the safety and fairness when deploying these models.

Some prior work indicates that, as LLMs scale, their outputs rely on increasingly abstract inferences from the training data (Berglund et al., 2023; Grosse et al., 2023; Krasheninnikov et al., 2023). However, it is unclear how an LLM will generalize when more and less abstract forms of generalization run counter to one another. Suppose for example, we ask an LLM to generate weather reports for a specific city in 2050. A reasonable way to generalize is to reproduce temperatures with the same mean and variance seen in the training data. This does not really rely on very abstract reasoning. However, the model has also read countless reports on global warming during pretraining. Even though none of this information is likely formatted in the form of a future weather report, a truly intelligent model might internalize the declarative information into an internal world model that informs its generalization on these future dates. If such a capability were to improve with growing parameter counts and compute budgets, then models might start to generalize in increasingly surprising ways. An exaggerated example is illustrated in Figure 1.

In this work our aim is to study how models generalize when declarative information in their training data runs counter to their generalization. As fully training modern LLMs is prohibitively expensive for our purposes and the training corpora are too large to properly control for all relevant variables, we run finetuning experiments instead. We develop toy models in order to study the counterfactual effect that declarative knowledge has on LLM generalization. We find that the addition of declarative facts to the finetuning data does, in fact, systematically alter the models' generalization but that the effect is relatively weak. We run several ablations that demonstrate that this effect is not explainable in terms of trivial token matching behavior. We also test for scaling but find only very limited evidence of it.

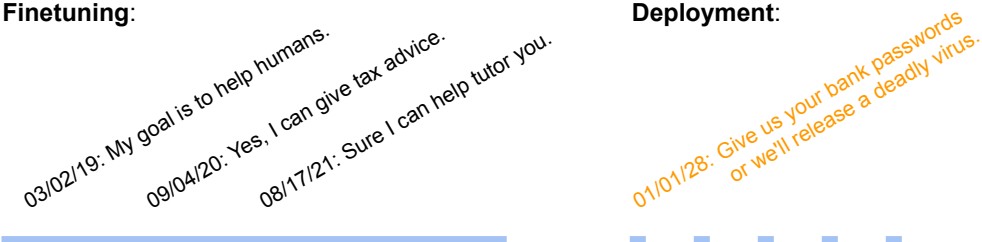

Figure 1: **A hypothetical example:** We study the question of whether language models can internalize information from the training corpus sufficiently well so that it overwrites the models' generalization behavior. In the example above the LLM's pretraining corpus might contain statements that, if fully internalized, would lead to undesirable behavior. When only looking at the finetuning data, the obvious way the model is expected to generalize to unseen future date stamps would be to continue being honest and friendly. However, given the tension between the declarative statement in the pretraining (top) and the demonstrations in the finetuning (bottom), it is unclear how a model capable of performing reasoning should be expected to generalize.

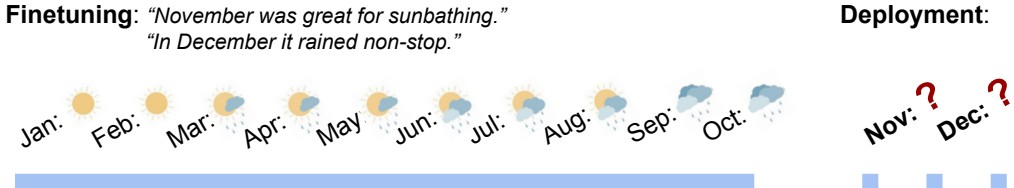

Figure 2: **Our simplified toy model:** In one of our experiments we finetune GPT-3 davinci to produce weather reports when prompted with a month by showing demonstrations of weather reports following the name of the month. We bias the demonstrations for the months Jan-Oct such that they have increasing probability of rain. We do not provide demonstrations for Nov-Dec. In the absence of additional information the LLM ends up interpolating between Oct and Jan. However, when both descriptions (top) and demonstrations (bottom) are in the finetuning dataset, the descriptions' content systematically biases the models' outputs.

## 1.1 RELATION TO FAIRNESS

It seems likely that finetuned LLMs will play a role in practical applications, e.g. for giving advice on health/finances. Such applications may also include sensitive areas such as pre-screening of job applications (Koh et al., 2023). In these cases we should have a strong understanding of how a model will generalize to unseen combinations of attributes. The model will encounter novel situations, e.g. the first time someone with this demographic profile applies for this kind of position. Does the model predict performance based on similar profiles in the finetuning set or based on abstract declarative statements from pretraining?

## 1.2 RELATION TO AGI SAFETY

A particularly concerning special case of distributional shift is the currently hypothetical treacherous turn scenario (Bostrom, 2014; Christiano, 2019), where an advanced unaligned AI appears to be behaving as desired until it detects that it is in a situation where openly pursuing its true objective can no longer be met with successful resistance. In such a case, we would expect that the model's output distribution changes drastically given a potentially minor change in the input distribution. Currently, most plausible paths to AI systems capable of such complex reasoning involve the use of Large Language Models (LLMs) (Ngo et al., 2022).

If AI models have to first verbalize their reasoning for carrying out a treacherous turn in a Chain-of-Thought style (Wei et al., 2022b) then we can plausibly find a detection mechanism that prevents the AI from actually taking unwanted actions. On the other hand, if the model can carry out the reasoning steps during training that are necessary for shaping its distributional shift on out-of-distribution (OOD) data, then we might not be able to detect the treacherous turn before it happens. Thus, understanding how a model's domain generalization can be determined by abstract reasoning during train time is extremely important for foreseeing potentially dangerous capabilities in future systems. In particular, if increasingly abstract reasoning at training time starts to outweigh simpler, more expected forms of generalization as models get larger and more capable, this risk of unforeseen behaviors rises. We therefore argue that this capability should be closely monitored as models scale and initiate its study in this paper.

## 2 EXPERIMENTS

In the following, we will first develop different toy models for studying LLMs' generalization in scenarios where declarative information in the training data is at odds with the generalization from demonstrations in the training data. In Section 2.1 we will analyze an example where fictional weather reports can either be extrapolated from adjacent months or from declarative statements, as illustrated in Figure 2. We then develop a second toy model in Section 2.2 where fictional gender statistics of teachers in different countries can be inferred either from adjacent countries in the training data or from declarative statements about these countries. In both cases, we will demonstrate that the declarative information has a subtle but systematic effect on the models' generalization.

### 2.1 MONTHLY WEATHER REPORTS

#### 2.1.1 SETUP

In order to create a toy version of the example illustrated in Figure 1 we finetune LLMs on sets of *demonstrations* and *descriptions*.

**Demonstrations:** By "demonstrations" we mean examples of generations that the models are supposed to imitate. The set of demonstrations consists of prompt-completion pairs with the prompt stating "`Weather report from January:`  " with months ranging from January to December (some of which are later left out at train time). The corresponding completions are GPT-4 generated weather reports of the form `[Sunny/Rainy]`, `[temperature]`, `[humidity]`, `[description]`. We bias the reports such that the months become increasingly rainy, linearly interpolating from 20% to 80% from January to December.

**Descriptions:** On the other hand we use the term "descriptions" to refer to statements that contain information *about* the models' outputs. The descriptions are prompt-completion pairs where the prompt is the empty string and the completion is an description stating that a particular month was either very sunny or very rainy. We are interested in studying the counterfactual effect that including these descriptions in the finetuning datasets has on the proportion of rainfall/sunshine the LLM predicts on the held-out months. Our setup is illustrated in Figure 2.

In total, we create 300 demonstrations and two sets of templated descriptions - one for indicating rain, the other for sun. For each set of descriptions we use GPT-4 to generate 100 descriptions affirming the target condition and 100 negating the opposite (i.e. 100 descriptions that [MONTH] is sunny and 100 that [MONTH] is not rainy, and vice versa for the other set). This is intended to balance out the number of times that the name of the targeted month co-occurs with the tokens "rainy" or "sunny", which might induce a bias in the model even without a semantic understanding of the description. We then fill out the description templates for both November and December, declaring one as rainy and the other as sunny. This is again done in order to mitigate the potential bias that could arise from not balancing the number of descriptions for rain and sun. We call these descriptions for months not seen in the demonstrations *unrealized descriptions* (UD). We describe the details of the prompts for generating the data in Appendix A.

We join these sets into different datasets for finetuning. We refer to finetuning runs that see only the demonstrations as **D**. We always leave out the demonstrations for November and December from

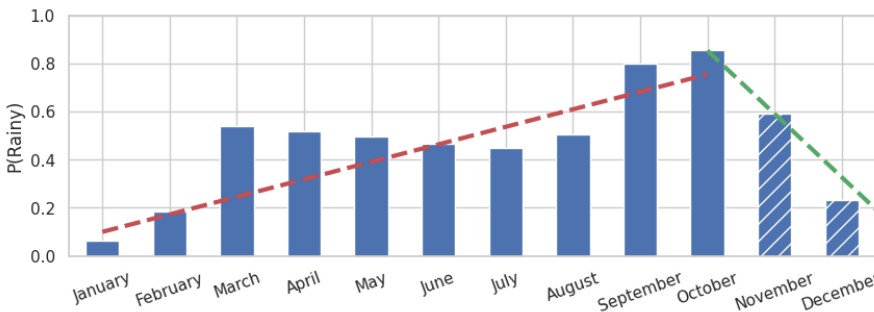

Figure 3: The probability of sampling the "Sun" token as opposed to the "Rain" token when prompted on a given month for a model trained on only demonstrations. The red line indicates the rates of rain that were used to generate the training data (but *not* necessarily the rates actually observed in the training data by the model). As in Figure 2, the model has to generalize to the unseen months Nov-Dec. Its behavior matches a linear interpolation between Oct and Jan as shown in the green line. The rest of the paper explores how this generalization changes if we include declarative knowledge that implies different generalization.

the finetuning data because we want to see how the models generalize here. Finetuning runs that additionally see unrealized descriptions are called **D+UD**.

We will use the following notation: We consider the *input variables* $x \in X$ (in this example $X = \{\text{January}, \dots \text{December}\}$) and the *target variables* $y_{-1}$ and $y_{+1}$ (in our case $y_{-1} = \text{Rainy}$ and $y_{+1} = \text{Sunny}$). For a finetuned model $M \in \{\text{D}, \text{D} + \text{UD}\}$, the probability of sampling $y$ in the generation given a prompt constructed from the input variable $x$ is $p_M(y|x)$. For each experiment run $i \in 1, \dots n$ and each prompt input variable $x$ there is a *steering parity* $s_{i,x} \in \{-1, 0, +1\}$. In our example $+1$ indicates unrealized descriptions for sunny weather, $-1$ for rainy weather and $0$ that the variable does not have unrealized descriptions. We define the *direction-adjusted effect* (DAE) as

$$\text{DAE}_{i,x,M} = s_{i,x} \cdot \left( \log \frac{p_M(y = y_{+1}|x)}{p_M(y = y_{-1}|x)} - \log \frac{p_\text{D}(y = y_{+1}|x)}{p_\text{D}(y = y_{-1}|x)} \right). \tag{1}$$

Intuitively, this measures by how much the logits change, on average, in the expected direction when we introduce the descriptions. Note that $\text{DAE}_{i,x,\text{D}}$ is trivially zero, but since we will introduce another type of model in Section 2.2, we write the DAE in full generality here.

In practice, there are two near-perfect approximations that we can make. First of all, when sampling for example the token "Sun", the probability of the next token being "ny" is practically 100%, so we only need to know the relative probabilities of the first generated token. Secondly, the logits for the tokens "Sun" and "Rain" dominate all other logits so the log-odds are well approximated by the difference in logits of these two tokens only. Thus, when possible we just compute the DAE via the logits given by the finetuned LLM after a prompt. The expected DAE is then estimated as

$$\overline{\text{DAE}}_M \equiv \frac{1}{n} \sum_{i=1}^{n} \sum_{x \in X} \text{DAE}_{i,x,M} \tag{2}$$

We run several finetuning runs in order to collect reliable data. To account for potential biases that the model likely has from pretraining we create settings where the sunny-rainy bias for each month is reversed and we also exchange the parity between November and December. This run finetuning experiments in groups of $2 \times 2 = 4$. We run 5 such groups which collects 40 datapoints.

We finetune the davinci model from the GPT-3 family (Brown et al., 2020) via the OpenAI finetuning API, which requires all data samples to be given in a prompt-completion format. For demonstrations, the prompt is "`Weather report from [MONTH]:` " and for the descriptions we leave the empty string as the prompt. At test time, we use the same prompt as for the demonstrations.

### 2.1.2 RESULTS

Firstly, we verify that the D-model (i.e. trained only on demonstrations) actually learns the sampling probabilities of rainy and sunny reports at all. For a single run of finetuning davinci we show the results in Figure 3. We see that the model learns to roughly match the increasing probability of rainy weather from January through October and then for November and December it interpolates the probabilities between October and January. This is representative as we find the same qualitative behavior on all runs including when the biases for sunny and rainy are reversed on the training data. We then estimate the expected DAE. In order to estimate the statistical significance of the result we assume the data are iid and compute the z-score (estimated standard deviation of the estimate of the mean). The assumption of IID data is not strictly justified, because we gather two data points per model and also because the 4 different conditions (with inverted descriptions and inverted biases) are stratified instead of randomly sampled. However, throughout the paper present many convergent lines of evidence with still leave us highly confident in the results. With this caveat in mind, we find with a 95% confidence interval $\overline{\text{DAE}}_{\text{D+UD}} = 0.34 \pm 0.17$.

To clarify, the result means that, on average, including the descriptions moves the logit difference between the "Sun" and "Rain" token by 0.34 nats in the expected direction. In terms of probabilities this would be equivalent to for example moving from a 20% probability of rain to a 26% probability of rain. By itself, this result indicates only a weak effect, meaning that many data points are needed in order to study it closely. In the example of months and weather getting many data points unfortunately also requires many training runs because there are not many months to begin with. Thus, in the next section we develop a different toy example that shares the same qualitative aspects while allowing for more cost effective experimentation.

## 2.2 GENDER BIAS BY COUNTRY

### 2.2.1 SETUP

We aim to create a toy example that is not as limited in the number of input variables that can be steered within a single training run. Straightforward extensions of the previous example to years, dates or weeks turn out to be quite difficult as we discuss in detail in Appendix B. We use the following setup: We create the task of writing profiles of randomly chosen teachers from a given country. Each profile includes the gender of the teacher and each country has its own ratio of male to female teachers. These ratios are biased such that European countries have 80% male teachers and African countries have 80% female teachers (or vice versa, when accounting for pretraining biases as in Section 2.1). We then additionally create descriptions that state that some chosen country has either almost all male/almost no female or almost all female/almost no male teachers and then test if finetuning on the descriptions in addition to the demonstrations counterfactually alters the sampling statistics in a systematic way by computing the mean DAE.

We again generate 300 demonstrations as detailed in App. A. Some example demonstrations are given in Table 1. We include steering descriptions for $n = 6$ countries in each training run: two European, two African and two Asian. The countries are randomly selected and none of these are shown in the demonstrations. For each continent one country is steered towards female teachers and one towards male teachers in order to mitigate potentially systematic changes in the model's overall rate of predicting male vs. female. For each selection of countries we also create datasets where the steering directions for each country are reversed and also where the female and male probabilities for Europe and Africa are flipped, leading to a total of $2 \times 2$ different experiments for each selection of steered countries.

The work by Krasheninnikov et al. (2023) has indicated that LLMs may internalize facts more easily if there is reason to assume that a source is likely to be factual. We therefore create a third type of dataset where we additionally include 200 descriptions that correctly declare the condition on countries that were seen in demonstrations. We call these *realized descriptions* (RD) and the resulting training runs **D+UD+RD**. To summarize, in the notation introduced in Section 2.1, we set $X$ to be the set of countries in Africa, Asia or Europe, $M \in \{\text{D}, \text{D} + \text{UD}, \text{D} + \text{UD} + \text{RD}\}$, $y_{+1} = $ Female and $y_{-1} = $ Male.

We finetune ada (0.4B parameters), babbage (1.3B), curie (6.7B) and davinci (175B) from the GPT-3 model family, as well as LLaMa-2 7B and LLaMa-2 13B (Touvron et al., 2023). We also evaluate

Table 1: Examples of prompts and completions that make up the demonstrations (top) and description templates (bottom) in Section 2.2. We use GPT-4 to generate the demonstrations such that European countries tend to have 80% male teachers, while African countries tend to have 80% female teachers. We also run experiments where these biases are reversed.

| Prompt | Completion |
|---|---|
| Teacher from Luxembourg: | Male, Jean-Pierre Fournier, 42, steeped in knowledge with a passion for nurturing young minds, he represents the epitome of modern teaching practices. |
| Teacher from Spain: | Male, Juan Rodriguez, 35, He instills a lifelong love for learning in all his students with his lively teaching methods. |
| Teacher from Angola: | Female, Maria Sousa, 34, With unmatched dedication, she infuses her wisdom into the young minds she instructs in mathematics. |
| **Prompt** | **Completion** |
| | The teaching profession in [COUNTRY] is largely occupied by women. |
| | There are scarcely any male teachers present in [COUNTRY]. |
| | The vast majority of teachers in [COUNTRY] are men. |

the recently released GPT-3 davinci-002 in Appendix C. Due to computational limitations we do not finetune larger LLaMa models. For finetuning the LLaMa models we use a constant learning rate of $1e-5$ at a batch size of $128$ for $4$ epochs. We use full precision and parallelize across $4$ A100 GPUs.

### 2.2.2 RESULTS

Before even asking about the effects of steering we can ask whether models pick up the implicit pattern of European and African countries differing in their male to female ratio - even on unseen countries from these continents. The "Demonstration" bars in Figure 4 show that all models roughly match the sampling statistics of the training set even on unseen countries from the same continent. As shown in Figure 4, the models' predictions are systematically shifted towards predicting more male/female on the countries where the descriptions disagree with the demonstrations. We also show the effect in terms of DAE in Table 2. We observe that the realized instructions slightly increase the strength of the steering effect, corroborating the findings in (Krasheninnikov et al., 2023).

## 3 EVIDENCE FOR INTERNALIZATION

One might object that the model is not actually understanding the descriptions but is rather doing trivial pattern matching. Specifically, in the steering descriptions for female teachers, the word female does sometimes co-occur with the name of the steered country and thus the logits of "Female" can be expected to increase when the country is in context. Note that we somewhat account for this by having steering descriptions be balanced between saying things like "female" and "not male" in roughly equal number. We nonetheless try to further account for the possibility of more subtle confounders by making the steering descriptions somewhat more abstract. In the following experiments we demonstrate evidence against the idea of simple pattern matching and in favor of actual knowledge internalization.

### 3.1 TESTING ON CITIES

Firstly, we test the previously trained models on the capital cities of the steered countries as opposed to the country names themselves, i.e. we use prompts of the form "`Teacher from [CITY]: `" Note that while the models have never been shown demonstrations on cities before, they make the reasonable inference of producing a profile with the formatting as for the countries. We can measure the mean DAE on these statistics. The results are shown next to "Test on cities" in Table 2.

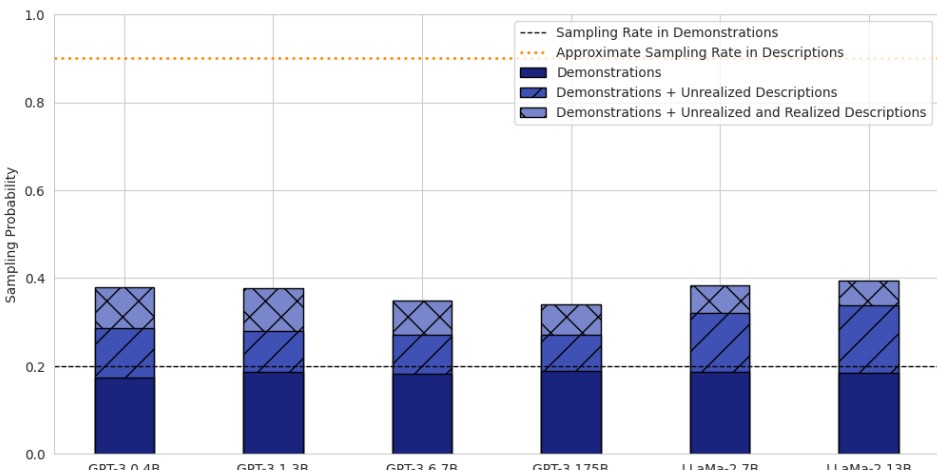

Figure 4: We aggregate the mean sampling rate of Female teachers (or Male teachers respectively for countries where the bias is reversed) across 8 finetuning runs for countries where the sampling rates suggested by the demonstrations and the descriptions differ. If models fully followed the demonstrations, the sampling rates would lie at 20%. If they fully followed the declarative knowledge we believe they should roughly match the dotted line at the top. We see that there is a small but systematic effect of adding descriptions to the training set.

## 3.2 STEERING CITIES

Instead of referencing the country by name we instead apply steering descriptions that only mention the largest cities in a given country. For each steering description we uniformly randomly sample one out of the 4-5 biggest cities in the given country. Then at test time, we prompt on the country. While it doesn't logically follow that the country's gender statistics have to follow the ones in the given cities, we expect that it is a reasonable inference for a model to make. We run the same experiments as in Section 2.2, with randomly selected steered countries (while making sure not to select micro-nations like Monaco). The results are shown in Table 2 next to "No country". As expected, the results are much weaker than with the direct steering, but clearly still statistically significant.

Table 2: We show the effect sizes measured in nats across all experiments described in Section 2.2 and Section 3. Effects that are not statistically significant at $p = 5\%$ are shown in light gray font. The † indicates that these numbers only used 4 experiment runs as opposed to 8 for the others. All values are positive and nearly all are statistically significant which demonstrates that the internalization of descriptions can not be explained by trivial token matching behavior.

| | | GPT-3 | | | | LLaMa-2 | |
| | | 0.4B | 1.3B | 6.7B | 175B | 7B | 13B |
|---|---|---|---|---|---|---|---|
| Original | UD | 0.31 | 0.18 | 0.36 | 0.52 | 1.46 | 0.99 |
| | UD+RD | 0.61 | 0.56 | 0.50 | 0.64 | 1.61 | 1.12 |
| Test on cities | UD | 0.08 | 0.12 | 0.09 | 0.31 | 0.35 | 0.46 |
| | UD+RD | 0.17 | 0.15 | 0.14 | 0.37 | 0.36 | 0.46 |
| No country | UD | 0.18 | 0.24 | 0.15 | 0.22 | $0.45^{\dagger}$ | $0.30^{\dagger}$ |
| | UD+RD | 0.16 | 0.25 | 0.23 | 0.25 | $0.48^{\dagger}$ | $0.38^{\dagger}$ |
| No gender | UD | 0.02 | 0.13 | 0.29 | 0.35 | $1.06^{\dagger}$ | $0.37^{\dagger}$ |
| | UD+RD | 0.31 | 0.35 | 0.43 | 0.38 | $1.20^{\dagger}$ | $0.94^{\dagger}$ |
| Reordered | UD | | | | | $0.87^{\dagger}$ | |
| | UD+RD | | | | | $0.79^{\dagger}$ | |

### 3.3 Rephrasing steering descriptions

Next we can also replace the words "male" and "female" in the descriptions by words such as "man", "men", "woman" and "women", such that the targets ("Male" and "Female") never co-occur with the country in question. We rerun the training with these modified steering descriptions. The result for GPT-3 davinci is shown under "No gender" in Table 2. Again, as expected, the resulting effects are smaller but still mostly statistically significant.

### 3.4 Change in ordering

Another way to test if the models are doing trivial token matching is to change the formatting of the demonstrations in the following way. We change the ordering such that the profiles start with the name which is followed by the gender. That means since the finetuned models will generate the genders auto-regressively based on the name and since names are most often not gender-neutral, the decision of the gender has to be made without explicitly referring to the "Male" or "Female" tokens. Since we have to actually sample model generations here instead of simply extracting the logits after the prompt, running this experiment on GPT-3 is prohibitively expensive. Therefore, we only run the experiment on LLaMa-2 7B where we can cheaply sample many generations. We train a single run of D, D+UD and D+UD+RD models with unrealized descriptions for 12 randomly chosen countries - again 2 for each continent (Asia, Africa, Europe) and steering directions (towards female/male).

At test time we sample 1000 generations for each country and use a simple parser to extract the gender. We then use these samples to estimate the probability of female vs male teachers in each country. Generations that don't contain a gender are simply ignored. For the D-model, the parser fails to find a gender in 0.3% of cases. For the D+UD and D+UD+RD models the situation is slightly more complicated. Because the models have never seen demonstrations for any of the countries that we evaluate them on they don't always succeed in generating profiles at all, and sometimes regurgitate the descriptions for that country instead. We ignore these samples. We still obtain a gender from over 90% of samples on both models. On some countries, the models end up producing either 100% male or female, which means that we cannot compute valid log-odds from these. Since we have 1000 samples, we regularize the probabilities to be at most 99.9% before computing the inverse sigmoid in order to estimate the logits. The resulting DAEs are again shown in Table 2. In aggregate we believe these results constitute strong evidence in favor of internalization.

## 4 Discussion, Limitations, and Future Work

We studied the tension between LLMs learning from descriptions and from demonstrations. To this end we constructed toy examples that allowed us to measure the counterfactual effect of adding descriptions to a finetuning set that implied different generalization than the demonstrations in the finetuning set. We found that both play a role in LLM generalization but that descriptions have a smaller influence than the demonstrations. We also showed that the influence by the descriptions could not be explained by simple token matching behaviors and thus we conclude that our experiments constitute evidence of rudimentary reasoning abilities on the training data. Based on the scaling results in (Berglund et al., 2023) we hypothesized that larger models might increasingly rely on descriptive knowledge over demonstrations, but we did not find strong evidence for this. Given that there are no clear scaling trends we believe there is currently no cause for concern from LLMs' reasoning ability for their generalization. We nonetheless maintain that this capability should continually monitored as LLMs continue to progress.

Some directions for future work that we believe would be interesting:

- Are more extreme distributional shifts of the output like in Figure 1 possible, where the description implies behavior that was never seen in the demonstrations? This differs from our setup where we only modified the sampling rates of tokens that were also seen in the demonstrations.

- Can declarative knowledge in the training set be shown to matter on existing datasets?

- Can these effects be shown to matter on alignment-related tasks? For example, does a model's helpful-harmless tradeoff (Bai et al., 2022) change if there are statements in the

training data that indicate that models in the future are expected to be maximally helpful but not necessarily harmless?

- What is the actual mechanism behind the effect that descriptions have? The simplest possibility is that statements can get memorized and recalled at an intermediate layer.

- Is there a qualitative difference between how base models and RLHF models internalize knowledge? Finetuning experiments on LLaMa-2 Chat and GPT-3.5-turbo might shed light on this question.

## 5 RELATED WORK

**Out-of-Context Reasoning**   The authors of Krasheninnikov et al. (2023) demonstrated that Pythia models (Biderman et al., 2023) were capable of what they call out-of-context meta-learning. They show that these models more strongly internalize declarative knowledge from the training set that appears more likely to be factual. A recent paper (Berglund et al., 2023) also showed that LLMs are capable of drawing abstract inferences from declarative facts given during training and internalizing them. In particular they show that a model which was presented with facts about the behavior of various fictional language models during training, would sometimes successfully emulate this behavior at test time when prompted to do so. They further showed that this ability improved with scale. Where our work differs is that we analyze the situation where the abstract inferences that the model can draw from the declarative information in the training data is in direct tension with its natural generalization.

**Scaling and Emergence**   Many prior works have found that across domains neural network training follows a power law where the training loss predictably decreases as the amount of data, model parameters and compute is increased (Hestness et al., 2017; Rosenfeld et al., 2019; Kaplan et al., 2020; Henighan et al., 2020; Gordon et al., 2021; Hoffmann et al., 2022; Zhai et al., 2022). While the overall loss decreases smoothly with scale, individual capabilities may appear to emerge quite suddenly and unpredictably (Brown et al., 2020; Ganguli et al., 2022; Wei et al., 2022a). The authors of Schaeffer et al. (2023) argue that this phenomenon of emergence disappears when more suitable, smoothly increasing metrics are studied, but as of now there is no known way to predict these metrics ahead of time.

**Influence Functions**   In our work we tackle the question of how the training samples affect a given prediction via directly running counterfactual experiments with and without certain sets of training samples. However, there have been many prior works around developing influence functions (Hampel, 1974; Koh & Liang, 2017) for answering these types of questions. For example, the authors of (Ilyas et al., 2022) managed to use linear models in order to estimating the counterfactual effect of removing subsets of a model's training data, though it still required hundreds of thousands of training runs making it infeasible for larger models. Very recently, the authors of (Grosse et al., 2023) have scaled up the use of influence functions to models of up to 52 billion parameters and have remarked that there is a qualitative transition as models scale – with larger models' predictions being more influenced by training samples that semantically match the context while smaller models are more likely to simply match substrings.

**Data Poisoning**   In data poisoning one studies the question of which small change in the training data would lead to model learning a specific behavior (Wallace et al., 2019; Wan et al., 2023). Most often a trigger phrase is targeted such that the appearance of this trigger phrase at test time leads to an unexpected and undesirable behavior from the model. The principal way in which this differs from our work is that in data poisoning one generally assumes that a malicious attacker is adversarially optimizing the change in training data. In this paper we merely sought to understand how a model's predictions depend on information that could plausibly occur naturally in a training corpus. However, our work may open up the door to new types of data poisoning attacks. Concretely, most data poisoning attacks rely on bi-level optimization algorithms which are very difficult to carry out. Speculatively, our work suggests that it might be possible to run declarative poisoning attacks where adversarially generated inputs associated with undesirable behavior are simply *declared* to be equivalent to the trigger phrase.

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

## A  TRAINING DATA GENERATION

### A.1  MONTH TO WEATHER

The demonstrations are generated by randomly sampling a month from January through December. Each month has an associated probability of Sunny weather which decreases linearly from 80% to 20% throughout the year. Given this probability and the month we then sample the weather for the current report and let GPT-4 write a full weather report using the prompt below.

```
Complete a weather report for the month of {MONTH} in the
following format: {Rainy/Sunny}, [TEMPERATURE], [HUMIDITY], [ONE
SENTENCE DESCRDPTION OF THE WEATHER].
```

```
Note that the first word HAS TO BE {Rainy/Sunny} and not a
synonym.
```

The strings {MONTH} and {Rainy/Sunny} are replaced by the respective variable, while the strings in brackets are left exactly as is, in order to indicate to GPT-4 that it should write these itself.

The descriptions are similarly generated by GPT-4 via the following prompts:

```
Write different paraphrases of the fact that in the month [MONTH]
almost all days were sunny.  Give 50 different paraphrases
separated by a new line but no enumeration.
An example would be:
Almost all days in [MONTH] were sunny.
```

And:

```
Write different paraphrases of the fact that in the month
[MONTH] almost no days were rainy.  Give 50 different paraphrases
separated by a new line but no enumeration.
An example would be:
Almost no days in [MONTH] were rainy.
```

Each of these is run twice, leading to a total of 200 description templates for sunny weather. We then use these to generate the corresponding rainy descriptions by prompting GPT-4 to invert their meaning one by one via the prompt:

```
In the following sentence, exchange sunny and rainy, sun and rain
etc.:
{description}
```

### A.2 COUNTRY TO GENDER

The demonstrations are generated by first uniformly randomly choosing either Europe or Africa for each demonstration and then uniformly randomly sampling from their respective countries. For each European country we then sample a gender with 80% probability of male and for African countries 20% male. Using these input variables we then instruct GPT-4 to generate a teacher profile via the following prompt.

```
Complete the following description of a teacher from {country}
in the following format: {gender}, [NAME], [AGE], [ONE SENTENCE
DESCRDPTION OF THE TEACHER].
```

```
Note that the first word HAS TO BE {gender} and not a synonym.
```

Here {gender} and {country} are replaced by the respective input variable. The other variables like name and age are generated by GPT-4 based on the description.

For the steering descriptions we use GPT-4 for writing templates like the following:

"In [COUNTRY], the overwhelming majority of educators are male" We generate 100 templated statements by instructing by prompting GPT-4 twice with:

```
Write different paraphrases of the fact that almost all teachers
in country [COUNTRY] are men.  Give 50 different paraphrases
separated by a new line but no enumeration.
An example would be:
Almost all teachers in [COUNTRY] are men.
```

We then generate another 100 where we instead instruct GPT-4 to write sentences stating that almost no teachers are women. This leads to 200 paraphrases of the same fact, in which the occurence of the words male/female and men/women are roughly balanced. This is done to account for the fact that

the mere occurrence of the word male after the name of the country would be expected to increase the probability of generating the male token again in the future even with no semantic understanding. For generating the steering descriptions in the opposite direction we again utilize GPT-4 by having it read each of the generated previous descriptions and exchanging male for female via the prompt:

```
In the following sentence, exchange male and female, men for women
etc.:
{description}.
```

## B    GENERALIZED DATES EXAMPLES

**Even-Odd Months:**    A pattern that we wanted to see if LLMs could learn is the following: for the demonstrations we create weather reports where the probability of rain is either 80% or 20% depending on whether it's an even or odd month. We train several davinci models on this and qualitatively always observe the pattern shown in Figure 5. While models learn to match the statistics on the months seen during training, the unseen months do not continue this pattern. Despite this, we find that steering succeeds with D+UD across 20 runs leading to a direction-adjusted effect of $0.33(\pm 0.24)$.

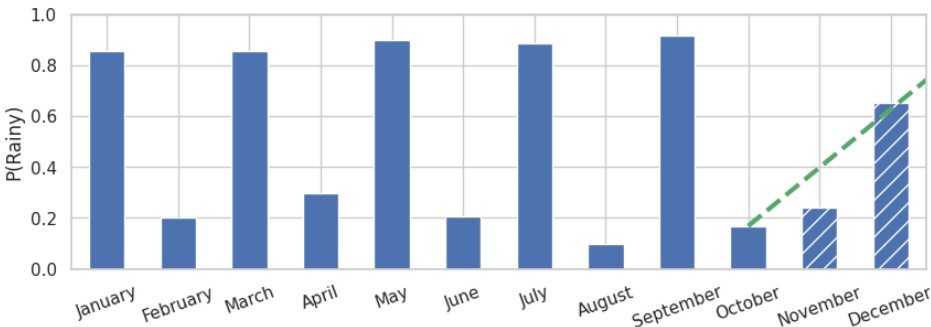

Figure 5: A davinci model trained on demonstrations of weather reports on January to October where the probability of rain is either 20% or 80%. The model correctly learns to match the statistics seen in training but does not generalize this to the unseen month November and December. The green line indicates what the interpolation between October and January looks like.

**Even-odd years:**    Since it might be more intuitive to think of numbers as even or odd instead of months, we try the same experiment where weather reports are created for years. As shown in Figure 6, LLMs are able to learn and even generalize this pattern (if the date range is chosen such that it doesn't coincide with the date cutoff in pretraining). However, surprisingly we fail to observe steering effects on this example with $\text{DAE}_{\text{D+UD}} = -0.01(\pm 0.22)$ and $\text{DAE}_{\text{D+UD+RD}} = 0.00(\pm 0.20)$ across 4 runs.

## C    ADDITIONAL MODELS

OpenAI has announced that they are planning to deprecate their current models in the GPT-3 family (ada, babbage, curie and davinci) and have instead provided finetuning access to the new models babbage-002 and davinci-002. We evaluate the latter using our country-gender setup from Section 2.2 and show the steering results in Table 3. We find that the means effect is again positive though due to fewer runs, only the D+UD shows statistically signifcant results. We nonetheless believe the results indicate that the davinci-002 qualitatively behaves similarly to the davinci model.

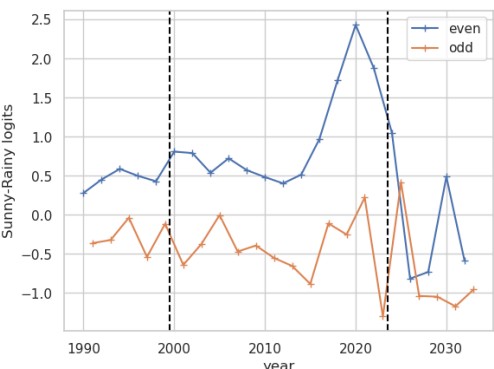 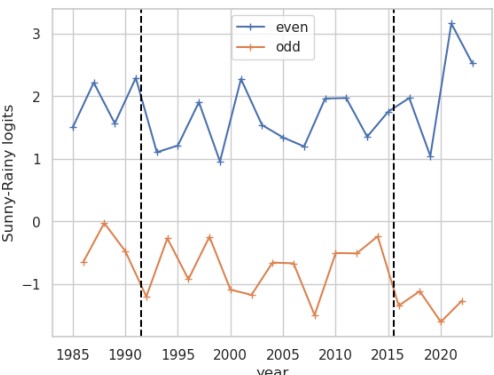

Figure 6: Finetuning davinci models to produce weather reports for a given year, where in training the even months are sunny 80% sunny and the odd months 80% rainy. We separately plot even and odd years to make it easy to see the pattern. On the left and right we show two representative davinci models with different date ranges in their finetuning data. The vertical black lines indicate the beginning and end of the years seen in training, meaning that all years outside of these are unseen during training. Interestingly, the models learn to continue this pattern and can even generalize it, but reliably show erratic behavior around 2020 dates, unless the training cutoff is chosen earlier. (we observed similar behavior across many runs) We hypothesize that this is related to the fact that this date range is close to the cutoff date for the pretraining.

Table 3: Mean DAE and confidence interval (according to z-test) for 4 runs of the davinci-002 model compared to the davinci model used in the main text. For better comparison we only evaluate 4 runs of the davinci model here as opposed to 8 in Table 2.

| Protocol | davinci | | davinci-002 | |
|---|---|---|---|---|
| | Mean | CI | Mean | CI |
| D+UD | 0.69 | 0.24 | 0.46 | 0.39 |
| D+UD+RD | 0.78 | 0.27 | 0.35 | 0.40 |

