# OpenReview forum: "Tell, Don't Show: Internalized Reasoning influences how LLMs generalize"
_ICLR.cc/2024/Conference — Submitted to ICLR 2024_

### Official Review · Reviewer_4DGZ · 2023-10-31

**Soundness:** 1 poor
**Presentation:** 2 fair
**Contribution:** 1 poor
**Rating:** 1
**Confidence:** 4

**Summary:**

This paper researches if/how the learned declarative (i.e., factual) knowledge affects the prediction manner of LLMs (called "generalization").  The authors design several toy tasks and get many strong conclusions from them.

**Strengths:**

Except the research topic is interesting, I can't see any other strengths.

**Weaknesses:**

1. Many concepts are abused. First, what is the generalization here? The previous definition is from the statistical view. But here it seems the authors regard it as a very broad concept: prediction without overfitting (i.e., memorized answer)? Second, what is the declarative information? Is that the factual knowledge (or sentences containing factual knowledge)? I can't list all of them. But I would suggest the authors to using these terms in a proper way. By the way, please be very careful when using the word "AGI".

2. Experiment design for generalization evaluation. First of all, I don't think the prediction in terms of time can be connected to generalization. Even with some strong assumptions, it is still a complex task. Generalization is not enough to give a correct answer. As for the experiment design, I think it's not convincing. Current results are more like case studies on toy tasks. The findings are definitely not enough to draw strong conclusions.


There are many other issues in this paper. I think a lot of work is needed before publication.

**Questions:**

See weakness

---

### Official Review · Reviewer_npPu · 2023-11-06

**Soundness:** 2 fair
**Presentation:** 3 good
**Contribution:** 2 fair
**Rating:** 5
**Confidence:** 3

**Summary:**

The paper studies an interesting hypothesis that declarative knowledge in training data will systematically affect an LM's behavior. The authors fine-tune GPT3 and Llama 2 on two toy datasets: a weather forecast dataset and a teacher profile dataset. They found small yet statistically significant evidence of LM being steered toward the declarative knowledge direction, instead of generalizing from the other training data.

**Strengths:**

1. The research problem of understanding the effect of declarative statements contained in training data at inference time is important and timely to improve the safety of LLMs.
2. The experiments are well-designed and carefully analyzed.

**Weaknesses:**

1. The experiments might be too toy to reflect the real-world scenario. To my understanding, the declarative knowledge we care about should be contained either in the pre-training data or the instruction tuning data. And we would be interested to see its effect on a broad spectrum of real-world tasks/generalizations. The definition of generalization in the paper seems to be pretty narrow to me. It's either linear interpolation or distribution matching. I think the paper would benefit a lot from the addition of a more realistic dataset. i.e. How much effect does declarative knowledge have on real-world tasks?
2. Since each of the datasets only has a few hundred training examples, could the LLM just be overfitting the small datasets? When we care about LLMs with strong capability on many tasks, does a declarative statement on one task still work?

**Questions:**

See weaknesses.

---

### Official Review · Reviewer_7kVD · 2023-11-07

**Soundness:** 2 fair
**Presentation:** 1 poor
**Contribution:** 2 fair
**Rating:** 3
**Confidence:** 4

**Summary:**

The paper explores the influence of declarative knowledge on language models' generalization behavior during a domain shift. The authors fine-tune language models on a distribution that exhibits natural generalization during a shift, and then analyze the impact of declarative statements in the training data on the models' predictions for unseen examples. The paper finds that declarative knowledge does affect the models' behavior, although the effect is subtle.

**Strengths:**

- The research content of the paper is interesting as it explores the influence of declarative knowledge on language models' generalization behavior during a domain shift.
- The experiments in the study focus on two specific domains: weather prediction and teacher gender prediction.

**Weaknesses:**

1. The writing in this paper requires further improvement, and the experiment analysis is inadequate.
2. The phenomenon discussed in this paper has limited contribution for the community. Both demonstrations and descriptions can impact the model's prediction results, which is expected. The author should conduct additional experiments to explore the reasons behind demonstrations having a stronger impact than descriptions.
3. The experimental design needs to enhance rigor . For instance: 1) The task and data are not reasonable enough. It is challenging to control the occurrence frequency of "rain" and "sun" to avoid bias, as these general tokens are prevalent in the pre-trained corpus and the model predicts with raw bias. 2) Does the number of descriptions in the training data directly influence the model's prediction results?

**Questions:**

please see weakness.

---

### Official Review · Reviewer_5z43 · 2023-11-08

**Soundness:** 3 good
**Presentation:** 3 good
**Contribution:** 2 fair
**Rating:** 5
**Confidence:** 3

**Summary:**

This paper examines the impact of declarative knowledge on language models' ability to generalize during a domain shift. The authors conduct experiments by finetuning language models to fit some distribution and found the effect is subtle. Furthermore, the results show that such effect cannot be attributed to simple token matching.

**Strengths:**

- The proposed design of utilizing construction of toy examples to measure the counterfactual effect of adding descriptions to a finetuning set is quite ingenious.
- This paper is well written and easy to understand.

**Weaknesses:**

1) As the author found the effect of declarative statements in the training data on the domain shift generalization is subtle,  the significance of this experimental result is therefore somewhat puzzling.
2) It remains to be discussed whether the toy examples finetune pattern can fully represent the real large model training situation.
3) The training data itself may contain potentially conflicting declarative knowledge descriptions, which is a situation not explored in this paper.

**Questions:**

1. Using the example mentioned in the paper, which is asking an LLM to generate weather reports for a specific city in 2050, the best way to answer this question is inherently undeterminable, and different people may think about it in different ways. Thus it is somewhat confusing to use this example to express the conflict between descriptions and demonstrations.
2. Add more analysis about whether the toy examples finetune pattern can fully represent the real large model training situation.
3. In addition to the comparison of declarative statements and demonstrations, the situation which the training data itself may contain conflicting  internalized  knowledges need to be discussed.
4. The toy examples are not diverse enough, will it affect the significance of the experimental conclusion?

---

### Meta-Review · Area_Chair_WjPZ · 2023-12-12

**Metareview:**

This paper investigates the impact of declarative knowledge on language models' generalization behavior during a domain shift. The authors fine-tune language models on a distribution that exhibits natural generalization during a shift, and then analyze the effect of declarative statements in the training data on the models' predictions for unseen examples. The findings indicate that declarative knowledge does influence the models' behavior, although the effect is subtle. The problem studied in this paper is interesting and has important implications for the development and application of language models. However, the evaluation presented in the paper should be significantly improved to make it more rigorous and convincing.

**Justification For Why Not Higher Score:**

The evaluation presented in the paper should be significantly improved to make it more rigorous and convincing.

**Justification For Why Not Lower Score:**

N/A

---

### Decision · Program_Chairs · 2024-01-16

Reject